# Genetic Diversity and Structure Revealed by Genomic Microsatellite Markers of *Mytilus unguiculatus* in the Coast of China Sea

**DOI:** 10.3390/ani13101609

**Published:** 2023-05-11

**Authors:** Xuelian Wei, Zeqin Fu, Jiji Li, Yingying Ye

**Affiliations:** National Engineering Research Center for Marine Aquaculture, Zhejiang Ocean University, Zhoushan 316022, China; weixl1247@163.com (X.W.); zeqin@beike.cc (Z.F.); lijiji@zjou.edu.cn (J.L.)

**Keywords:** *Mytilus unguiculatus*, microsatellite markers, genetic diversity, genetic structure

## Abstract

**Simple Summary:**

Microsatellite markers are widely used in genetic breeding and population genetic structure analysis. In this study, microsatellite markers are used to analyze the genetic diversity and population genetic structure of *Mytilus unguiculatus* in seven coastal areas of China. The results showed that *M*. *unguiculatus* has high genetic diversity with genetic structure differences observed between the Qingdao population and the other six populations. The genetic structure of *M. unguiculatu* is relatively weak. These findings provide a molecular biological basis for the rational development and protection of wild germplasm resources of *Mytilus unguiculatus* in China and can serve as a data reference for the formulation of reasonable breeding programs.

**Abstract:**

The hard-shelled mussel *Mytilus unguiculatus* plays an important role in mussel aquaculture in China due to its characteristic and nutritive value. In this study, 10 microsatellite loci are used to study the genetic diversity and genetic structure of seven location populations of *M. unguiculatus* in coastal areas of China. The results of amplification and genotyping show that the observed heterozygosity (*H*_o_) and the expected heterozygosity (*H*_e_) are 0.61~0.71 and 0.72~0.83, respectively. *M. unguiculatus* has high genetic diversity. The inbreeding index (*F*_IS_) of *M. unguiculatus* is significantly positive (*F*_IS_: 0.14~0.19), indicating that inbreeding might exist within populations. The genetic structure of *M. unguiculatus* is weak within populations from the East China Sea All results showed that genetic differences existed between the Qingdao population from the Yellow Sea and other populations from the East China Sea. It does not detect a population bottleneck event or expansion event in the populations. The results from this study can be used to provide important insights in genetic management units and sustainable utilization of *M. unguiculatus* resources and provide a better understand of genetic structure of marine bivalve with similar planktonic larval stage in the China Sea.

## 1. Introduction

*Mytilus unguiculatus* (Mollusca; Bivalvia; Mytiloida; Mytilidae; *Mytilus*) is an offshore warm–temperate benthic shellfish. The shell of *M. unguiculatus* is tip and thin, and the shell surface is often brown. In East Asia, it is widely distributed in the littoral of the Japan Sea, Bohai Sea, Yellow Sea, East China Sea, and South China Sea [1,2]. In China, it has a long breeding history, and the main breeding area is in Zhoushan City, Zhejiang Province, where the number of breeding currently exceeds 6 billion grains per year [3,4]. Due to its high nutritional value and delicious taste, *M. unguiculatus* is widely consumed by people [5]. *M. unguiculatus* is one of the important aquaculture shellfish species in China. With the increasing demand of people, *M. unguiculatus* have become one of the key fishing targets. However, the generation replacement rate of *M. unguiculatus* is slow, and large-scale and high-intensity harvesting is bound to reduce mussel resources [6,7]. Therefore, studying the genetic structure of wild mussels can provide valuable information for improving the quality of provenances and cultured mussels, which can effectively protect resources of genetic variation in wild populations and improve the yield of *M. unguiculatus*.

Microsatellite DNA (Simple Sequence Repeats, SSR) is a molecular marker that is widely distributed in the genome of eucaryon. It consists of a repetitive DNA sequence with a certain number of base pairs (1 to 6 bp) [8,9]. Microsatellite DNA has a higher mutation rate than other regions of DNA [10]. There are different repeats of DNA motifs between individuals within species because of the influence of the environment or other factors. Therefore, this characteristic makes significant variation between individuals and populations, which means the microsatellite DNA is one kind of high polymorphism molecular marker [11]. Additionally, microsatellite DNA follows the Mendelian law and is inherited in a codominant manner, which makes it more powerful than previous markers in studying the relationship among alleles in an individual. It can also distinguish between homozygous and heterozygous alleles [12]. It is easy to obtain a microsatellite as it is ubiquitous in the genome. PCR can be used to obtain sequence details, which can then be analyzed with genetic statistical methods for further analysis. Therefore, microsatellite markers are used in many fields, including genetic breeding [13], genetic resource status assessment [14,15], genetic linkage analysis [16,17], and genetic mapping [18,19].

Currently, the population genetics of *M. unguiculatus* are mainly studied using mitochondrial molecular markers [20,21,22]. There are few reports of using SSR to analyze *M. unguiculatus*. The objectives of this study were to investigate the genetic diversity and population genetic structure of *M. unguiculatus* using microsatellite markers and compare the results with those obtained using mitochondrial molecular markers. This study aims to provide information on the geographical distribution patterns and evolutionary processes of this species, as well as to fill knowledge gaps in related research fields. Additionally, the new data obtained will aid fishery resource managers in assessing present resources and developing conservation strategies.

## 2. Materials and Methods

### 2.1. Sample Collection and DNA Extraction

The seven sampling locations of *M. unguiculatus* crossed from the Yellow Sea to East China Sea (i.e., Qingdao, Zhoushan, Xiangshan, Yuhuan, Taishan, Pingtan, and Xiamen) (Table 1 and Figure 1). In total, 50 individuals were collected and analyzed from each population (Table 1, Figure 1). Specimens were collected from the wild environment rather than areas of aquaculture breeding programs. Adductor muscle fragments were removed from each mussel and stored in 95% ethanol and below −20 °C until DNA extractions. The DNA extraction followed the method that was reported in Aljanabi and Martinez (1997) with slight modifications [23]. DNA quality was determined by electrophoresis in 1% agarose gel. Total DNA concentration was diluted to 30–40 ng/μL and stored at −20 °C for further analysis.

### 2.2. Microsatellite Analysis

All primers used in this study were selected from Fu et al. [24]. Ten pairs of primers were chosen based on their good polymorphism and high resolution, MC08, MC18, MC44, MC47, MC54, MC57, MC66, MC74, MC76, and MC99 (Table 2). Each primer was labelled with a fluorescent dye (FAM and HEX, Applied Biosystems, USA) at the 5’ end. PCR was performed in 25 µL containing template DNA (20~50 ng), forward primer (4 ρmol), reverse primer (4 ρmol), and 2 × Taq MasterMix (10 µL; CW0716; Cwbiotech., Beijing, China). The reaction procedure was pre-denaturation at 94 °C for 3 min, followed denaturation by 35 cycles at 94 °C for 30 s, 55~61 °C for 30 s (each pair of primer was screened in gradient temperature), extension at 72 °C for 40 s, and final extension at 72 °C for 10 min. For evaluating the amplifications, we used 8% non-denaturing polyacrylamide gel electrophoresis to detect PCR products. Then, the genotyping of PCR products by capillary electrophoresis was performed using the BiopticQsep100 DNA Analyzer in Shanghai Generay Biotech Co. Ltd. (Shanghai, China) with the following conditions: 1 µL PCR product, together with 9 µL HiDi-formamide (Applied Biosystems), and 1 µL GS500LIZ (P/N 4322682) size standard (Applied Biosystems).

Finally, capillary electrophoresis genotyping was performed using the BiopticQsep100 DNA Analyzer in Shanghai Generay Biotech Co. Ltd. (Shanghai, China) under the following conditions: 1 µL PCR product, 1 µL GS500LIZ (P/N 4322682) size standard (Applied Biosystems), and together with 9 µL HiDi-formamide (Applied Biosystems).

### 2.3. Genetic Data Analysis

After genotyping, the number of alleles (*N*_a_), observed (*H*_o_), and expected (*H*_e_) heterozygosity were evaluated using GenAlEx v6.501 [25]. The FSTAT v2.9.3 software [26] was used to estimate the allele richness (*A*_r_) by setting the minimum sample size to 46 and the genetics inbreeding index (*F*_IS_). The MICRO-SATELLITE ANALYSER (MSA) v4.05 was used to estimate the genetics differentiation index (*F*_ST_) [27]. The program GENPOP v4.5 [28] was used to test each marker for Hardy–Weinberg Equilibrium (*HWE*) and the *p* value. Arlequin v3.5 [29] was performed the term hierarchical analysis of molecular variance (AMOVA) with 10,000 permutation. Using program NeESTIMATOR v2.0 [30] to estimate the effective size of each population based on the linkage disequilibrium (each allele’s frequency > 0.05 CI = 95%). The program BOTTLENECK v1.2 [31,32] performed the bottleneck test to analyze whether there had been a population event (expansion or bottleneck). The setting was as follows: model = Infinite allele model (I.A.M.) [33], Stepwise mutation model (S.M.M.) [34] and Two-phase model of mutation (T.P.M.) [35] with 10% I.A.M., 90% S.M.M., and 10,000 permutations [36]. The Wilcoxon test was used to check whether there was significant excess heterozygote [37]. Finally, the software MICROCHECKER v2.2.3 [38] was performed to test for the presence of null alleles.

The program Populations v1.2 [39] was used to construct the *NJ* phylogenetic tree among seven populations based on the Nei’s standard genetics distance [40]. The software STRUCTURE v2.3 was performed the clustering analysis based on Bayesian method (K set 1 to 7, 20 runs, each run used an admixture model, MCMC = 1,000,000, burn-in = 25,000). The results were submitted to the online tool STRUCTURE HARVESTER [41] for evaluation of the best K value and the program CLUMPP [42] was used to plot the STRUCTURE analysis.

## 3. Results

### 3.1. Genetic Diversity

Results from the analysis of genetic diversity showed the average numbers of alleles (*N*_a_) ranged from 5.5 (TS) to 7.1 (YH). The observed and expected heterozygosity (*H*_o_ and *H*_e_) varied from 0.61 (QD) to 0.71 (YH), and 0.72 (QD) to 0.83 (YH). The inbreeding index (*F*_IS_) ranged from 0.14 (YH) to 0.19 (ZS, XS, and TS). All loci did not significantly deviate from *HWE* (Table 2).

### 3.2. Genetic Structure

Based on the geographic distribution of sea areas, *M. unguiculatus* were divided into two groups (Group 1: QD and Group 2: ZS, XS, YH, TS, PT and XM). Results of AMOVA showed that genetic differentiation among groups was over 4.1% of total variation, with 1.3% attributed to variation among populations within groups and the remaining 94.6% attributed to within populations (Table 3). The pairwise *F*_ST_ values revealed that Qingdao had a slight significant genetic divergence with the other six populations. In the six investigated populations from the East China Sea, most pairwise genetic distances based on fixation index (*F*_ST_) were not significant. The gene flow (*N*_m_) ranged from 3.80 to 833.08, indicating frequent gene exchange between populations at different sampling locations (Table 4). The Bayesian assignment analysis supported the result of pairwise *F*_ST_. Under the best K value (K = 2, Figure 2), Qingdao was divergence with other six populations (Figure 3). The phylogenetic tree also showed that the population of Qingdao was far from other populations (Figure 4).

### 3.3. Historical Dynamics

The *N*_e_ estimates for seven populations of *M. unguiculatus* were large, ranging from 694.5 on TS (CI = 207.6 − infinity) to infinity on QD (CI = 372.6 − infinity). In the bottleneck test, based on three different models, it was not significant that the heterozygote exceeded any other populations. The allele distribution conforms to the L-shaped distribution without bottleneck effect events. It indicates that no genetic bottleneck has recently occurred in the populations of *M. unguiculatus* along the coastline of the China Sea (Table 5). Furthermore, the mode-shift test also indicated no evidence for bottleneck events in the recent evolutionary history of *M. unguiculatus* (Figure 5).

## 4. Discussion

High genetic diversity generally enables species to adapt to environmental changes and avoid inbreeding. Microsatellite alleles and heterozygosity are crucial parameters to determine the genetic diversity of a population [43,44,45]. A population that exhibits heterozygosity greater than 0.5 is generally considered to have high genetic diversity [46]. In the present study, the *H*_o_ and *H*_e_ of *M. unguiculatus* were 0.61 to 0.71 and 0.72 to 0.83, respectively. This finding is consistent with previous studies by Liu et al. [20], Feng et al. [21], and Wei et al. [22] that investigated the same species in the same area using mitochondrial molecular markers. For other marine shellfish analyzed using microsatellite markers, *Meretrix meretrix* had an average *H*_o_ of 0.3878 and *H*_e_ of 0.7996 [43], *Crassostrea angulata* had an average *H*_o_ of 0.647 and *H*_e_ of 0.872 [47], and *Sinonovacula constricta* had an average *H*_o_ of 0.7525 and *H*_e_ of 0.6866 [48]. In comparison with the *H*_o_ and *H*_e_ values of *M. unguiculatus* obtained in this paper, the genetic diversity level of *M. unguiculatus* is slightly higher than that of *Meretrix meretrix*, which is close to that of *Crassostrea angulata* and *Sinonovacula constricta*, indicating a high level of genetic diversity. This may be related to the in vitro fertilization of *M. unguiculatus*, the ability to reproduce on a large scale, and the spread of larvae [21]. The results of genetic diversity showed that the current germplasm resources of *M. unguiculatus* were in good condition and had good breeding prospects.

The *F*_IS_ value is an important index to measure the genetic dynamics of the population, reflecting the balance between *H*_o_ and *H*_e_. When *F*_IS_ is positive, it indicates that the heterozygote is missing; when *F*_IS_ is negative, it indicates that the heterozygote is excessive [49,50]. The *F*_IS_ values of the seven populations in this study ranged from 0.14 to 0.19, and the *F*_IS_ values were significantly positive. The *F*_IS_ values estimated were between 0 and 1, which was consistent with the results that the *H*_o_ was less than the *H*_e_ of the study populations. This result indicates that there is a loss of heterozygosity in the mussel population. This indicates that there may be inbreeding in the population, and it may also be caused by rare base deletion caused by human interference and other factors [51,52].

The results of the genetic structure analysis and population comparison revealed a low level of genetic variation in Qingdao population and confirmed their distribution on areas of sea. The *F*_ST_ value is an important index to reflect the degree of population genetic differentiation [53]. Wright and Maxson [54] distinguished four levels of genetic differentiation, based on *F*_ST_ estimations, little differentiation (0 < *F*_ST_ < 0.05), moderate differentiation (0.05 < *F*_ST_ < 0.15), large differentiation (0.15 < *F*_ST_ < 0.25), and very large genetic differentiation (*F*_ST_ > 0.25). Therefore, the pairwise *F*_ST_ (0.003~0.065) was quite small for the populations studied. The pairwise *F*_ST_ values, including Qingdao population, were higher than others. The phylogenetic tree (Figure 4) showed that the genetic distance of Qingdao was far away from the other populations and the STRUCTURE analysis (Figure 3) also supported that it was divided into two groups based on the geographic distribution. The results of *N*_m_ can be confirmed by *F*_ST_ results. Relevant studies suggest that when *N*_m_ > 4, the effect of genetic drift can be ignored [55,56]. All populations are in a state of random mating, and gene flow is the main factor affecting population genetic differentiation. In this study, the *N*_m_ values of Qingdao population and the other six populations were all around 4, while the *N*_m_ values between other populations were far greater than 4. This shows that except for the Qingdao population, the other six populations have frequent gene exchanges and no obvious genetic differentiation. However, Wei et al. [22] used mitochondrial molecular markers to study *M. unguiculatus* in the same area. The results showed that the degree of genetic differentiation between the seven populations was not obvious, and no obvious genetic differentiation was found in the genetics of each population. It was caused by the origin and specificity of mtDNA. With genetic conservation of genes, researchers made it possible to use some specific mtDNA to gain mitochondrial DNA sequences/genome from unknown species [57]. However, the genetic convenience of mtDNA also limited the detection range and detection sensitivity.

The results of this study indicated that *M. unguiculatus* exhibits a weak genetic structure, possibly due to the influence of ocean currents. The China Sea has a complex system of ocean currents, including the Kuroshio current and its tributaries, the Coastal currents system (e.g., the northern Jiangsu current, the Zhejiang and Fujian current, and the South China Sea current), and local circulation, upwelling, and eddy currents (e.g., the central cold water mass in the Yellow Sea in summer and autumn, the northern cyclone vortex in the East China Sea, and the Yangtze dilute water) [58,59,60,61]. The influence of ocean currents on species formation and population genetic structure differentiation of marine organisms has long been confirmed [62]. Marine plankton have high passive dispersal ability, which can spread along the direction of ocean currents during the planktonic period, thus promoting genetic exchange between different populations [63,64,65]. The East China Sea is the main producing area of *M. unguiculatus* in China, and its mass breeding season is in winter every year. During this period, a large number of mussel larvae will be dispersed in the sea. *M. unguiculatus* with a long planktonic period (about 35d) [66] drift for a long distance under the impetus of marine currents, resulting in genetic exchanges and a weak genetic structure. The current artificial breeding technology for *M. unguiculatus* is relatively mature, and aquaculture practitioners typically obtain seedlings from the main producing areas and transport them to breeding sites. The aquaculture method involves using attached substrates for seedling culture in natural sea areas. This open breeding method leads to a large number of cultured sample larvae being dispersed in the sea during the breeding season and becoming mixed with wild mussel larvae. As a result, a weakening of the genetic structure of *M. unguiculatus* can occur due to the large number of cultured sample larvae mixing with wild mussel larvae and spreading with ocean currents.

The investigated seven populations of *M. unguicultas* did not show significant traces of bottleneck effects, indicating that crossbreeding between populations is likely the main factor contributing to the current level of genetic variation. Inbreeding among populations could result in the loss of genetic information, making it undetectable through molecular methods. Although an artificial seedling technique has been developed for farming *M. unguiculatus*, most of the seedling parents still originate from the natural producing area in the East China Sea. The increase in breeding demand has led to the embezzlement of natural habitats, resulting in a decline in natural samples and a reduction in genetic variation. At present, the breeding industry of *M. unguicultas* lacks natural population protection management, which can lead to the dispersal of a large number of cultured larvae into the sea during the breeding season. These larvae can mix with natural mussel larvae and weaken the genetic structure of *M. unguiculatus* in the wild population. Therefore, to conserve the germplasm resources of *M. unguiculatus*, we recommend reducing the genetic exchange between cultured and natural populations and minimizing the contamination of genetic information in natural populations.

## 5. Conclusions

Population genetics research on *M. unguiculatus* is critical in revealing its situation in the natural environment. In this study, seven *M. unguiculatus* populations in coastal areas of China were analyzed by microsatellite markers. The results showed that *M. unguiculatus* had high genetic diversity. Qingdao and other populations had genetic differentiation. The overall genetic structure was weak, and the presence and size of any bottleneck effect on genetic variation were difficult to assess. The genetic data generated in this study could provide insights into genetic management units and the utilization of *M. unguiculatus* resources.

## Figures and Tables

**Figure 1 animals-13-01609-f001:**
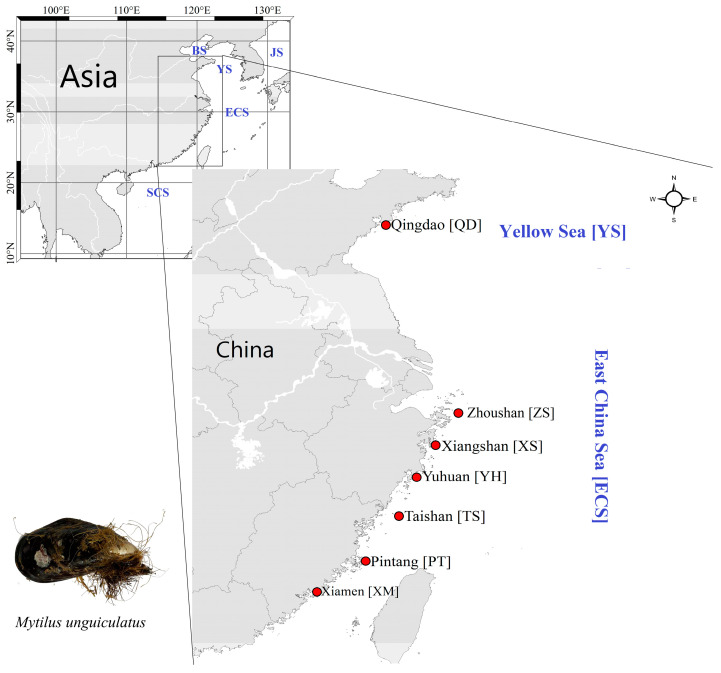
The map of sampling locations (BS: Bohai Sea; YS: Yellow Sea; ECS: East China Sea; SCS: South China Sea; JS: Japan Sea).

**Figure 2 animals-13-01609-f002:**
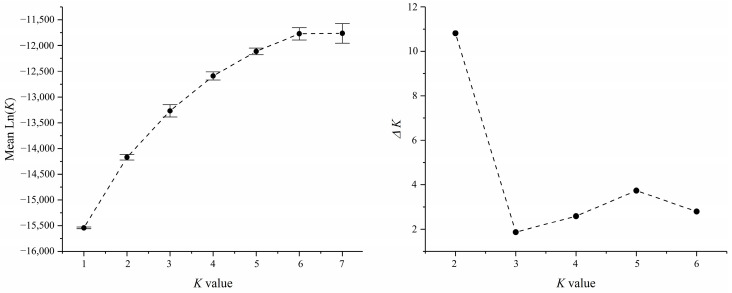
The STRUCTURE analysis of the *K* values, the error bars indicate standard deviation.

**Figure 3 animals-13-01609-f003:**
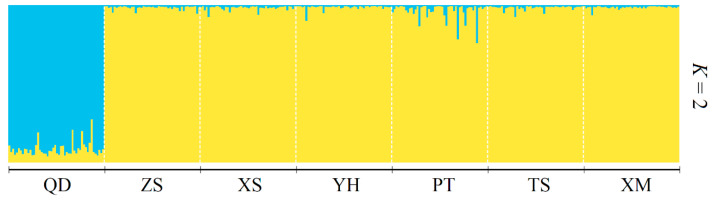
Bayesian individual assignment analysis carried out with STRUCTURE on seven populations of *Mytilus unguiculatus*.

**Figure 4 animals-13-01609-f004:**
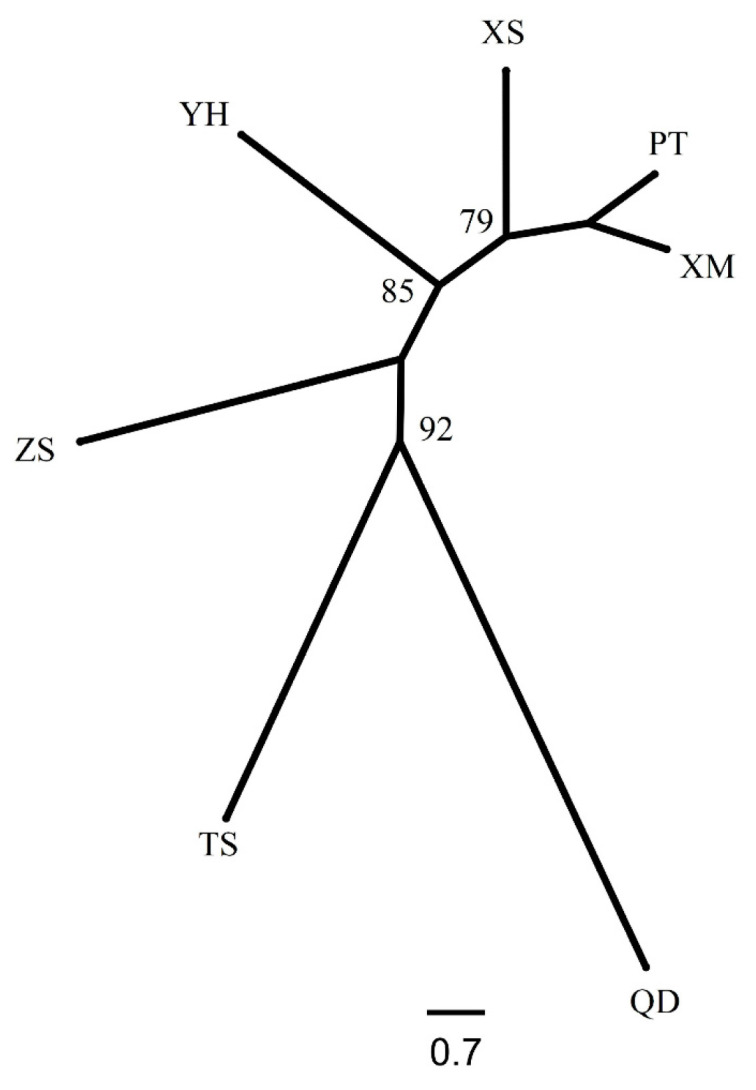
The *NJ* tree of seven populations of *M. unguiculatus*.

**Figure 5 animals-13-01609-f005:**
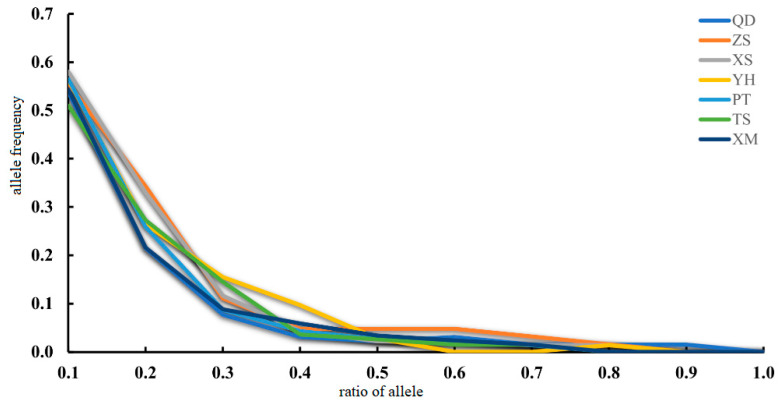
Allele frequency distribution of *M. unguiculatus*.

**Table 1 animals-13-01609-t001:** Sampling information of *M. unguiculatus*.

Sampling Site	Abbreviation	Coordinate	Sampling Date	Sample Size
Qingdao	QD	35°55′ N, 120°30′ E	September 2016	50
Zhoushan	ZS	30°12′ N, 122°42′ E	January 2017	50
Xiangshan	XS	29°14′ N, 121°58′ E	December 2016	50
Yuhuan	YH	28°14′ N, 121°24′ E	March 2017	50
Taishan	TS	26°57′ N, 120°47′ E	November 2016	50
Pingtan	PT	25°34′ N, 119°47′ E	February 2017	50
Xiamen	XM	24°35′ N, 118°23′ E	November 2016	50
Total				350

**Table 2 animals-13-01609-t002:** Genetic diversity in seven different sampling locations of *M. unguiculatus*.

Populations	Loci	MC08	MC18	MC44	MC47	MC54	MC57	MC66	MC74	MC76	MC99	Average
QD	*N* _a_	8	6	5	5	6	10	5	9	5	6	6.50
	*H* _e_	0.86	0.86	0.48	0.78	0.54	0.68	0.32	0.94	0.92	0.80	0.72
	*H* _o_	0.67	0.78	0.38	0.65	0.58	0.67	0.33	0.73	0.72	0.62	0.61
	*F* _IS_	0.23 *	0.09 *	0.21 *	0.17 *	−0.07	0.01 *	−0.03 *	0.22 **	0.22 *	0.23 *	0.15 *
ZS	*N* _a_	7	5	3	6	5	10	7	10	5	5	6.30
	*H* _e_	0.86	0.86	0.44	0.80	0.66	0.76	0.78	0.90	0.94	0.78	0.78
	*H* _o_	0.63	0.67	0.37	0.66	0.59	0.73	0.58	0.72	0.74	0.58	0.63
	*F* _IS_	0.27 *	0.22 *	0.16 *	0.17 *	0.11 *	0.04 *	0.26 *	0.20 *	0.21 *	0.26 **	0.19 *
XS	*N* _a_	10	4	3	6	6	6	12	9	6	7	6.90
	*H* _e_	0.98	0.80	0.50	0.86	0.62	0.70	0.84	0.94	0.90	0.74	0.79
	*H* _o_	0.68	0.65	0.38	0.66	0.60	0.62	0.75	0.79	0.70	0.55	0.64
	*F* _IS_	0.31 **	0.19 *	0.24 *	0.23 *	0.03 *	0.11 *	0.11 *	0.16 *	0.22 *	0.26 *	0.19 *
YH	*N* _a_	6	5	6	6	9	9	8	7	7	8	7.10
	*H* _e_	0.82	0.86	0.54	0.92	0.74	0.72	0.92	0.88	0.92	0.94	0.83
	*H* _o_	0.72	0.71	0.45	0.75	0.70	0.75	0.79	0.74	0.76	0.74	0.71
	*F* _IS_	0.12 *	0.17 *	0.17 *	0.18 *	0.05 *	−0.04	0.14 *	0.16 *	0.17 *	0.21 *	0.14 *
TS	*N* _a_	6	5	5	5	4	6	3	8	6	7	5.50
	*H* _e_	0.80	0.96	0.82	0.70	0.64	0.76	0.58	0.92	0.92	0.84	0.79
	*H* _o_	0.58	0.77	0.62	0.64	0.51	0.71	0.54	0.71	0.76	0.67	0.65
	*F* _IS_	0.31 **	0.19 *	0.24 *	0.23 *	0.03 *	0.11 *	0.11 **	0.16 *	0.22 *	0.26 *	0.19 *
PT	*N* _a_	6	6	6	7	6	8	10	9	5	6	6.90
	*H* _e_	0.82	0.88	0.62	0.76	0.68	0.60	0.90	0.92	0.90	0.88	0.80
	*H* _o_	0.62	0.74	0.47	0.67	0.59	0.68	0.69	0.76	0.70	0.65	0.66
	*F* _IS_	0.24 *	0.16 *	0.24 *	0.12 *	0.13 *	−0.13	0.23 *	0.17 **	0.22 *	0.26 *	0.18 *
XM	*N* _a_	5	5	6	6	6	7	10	10	6	7	6.80
	*H* _e_	0.84	0.90	0.58	0.72	0.72	0.58	0.96	0.88	0.90	0.82	0.79
	*H* _o_	0.68	0.74	0.47	0.68	0.66	0.66	0.70	0.76	0.71	0.63	0.67
	*F* _IS_	0.19 *	0.18 *	0.19 *	0.06 *	0.08 *	−0.14	0.27 **	0.14 *	0.21 *	0.23 *	0.15 *

*N*_a_: number of alleles; *H*_o_: observed heterozygosity; *H*_e_: expected heterozygosity; *F*_IS_: genetic inbreeding index; *: *p* < 0.05; **: *p* < 0.01.

**Table 3 animals-13-01609-t003:** The AMOVA analysis of genetic structure of *M. unguiculatus*.

Grouping	Sources of Variation	Degree of Freedom	Sum of Square	*F*—Statistic	% of Variation	*p*-Value
1: QD; 2: ZS, XS, YH, TS, PT, XM	Among groups	1	31.99	*F*_CT_ = 0.041	4.1	0.04
	Among populations within groups	5	38.73	*F*_SC_ = 0.013	1.3	0.001
	Within populations	693	2284.45	*F*_ST_ = 0.053	94.6	0.001

**Table 4 animals-13-01609-t004:** The pairwise *F*_ST_ (below diagonal) and *N*_m_ (above diagonal) between seven populations.

Population	QD	ZS	XS	YH	PT	TS	XM
QD	-	3.80	4.12	3.58	5.07	4.87	4.83
ZS	0.062 *	-	10.15	11.42	9.29	18.78	14.79
XS	0.057 *	0.024	-	16.87	11.37	208.08	110.37
YH	0.065 *	0.021 *	0.015 *	-	11.74	29.80	78.86
PT	0.047 *	0.026	0.022 *	0.021 *	-	28.96	17.58
TS	0.049 *	0.013	0.001	0.008	0.009	-	833.08
XM	0.049 *	0.017	0.002	0.003	0.014 *	0.003	-

Values with * indicate statistical significant difference from zero (*p* < 0.002, *p*-values adjusted for multiple comparisons using Bonferroni correction = 0.05/21).

**Table 5 animals-13-01609-t005:** The *N*_e_ estimation and Bottleneck test of *M. unguiculatus*.

Population	*N*_e_ Estimation	CI = 95%		Bottleneck Test			Mode-Shift
		Lower	Upper	I.A.M.	S.M.M.	T.P.M.	
QD	infinity	372.6	infinity	0.586	0.732	0.707	L-shaped
ZS	1365.6	532.7	infinity	0.565	0.709	0.683	L-shaped
XS	infinity	136.8	infinity	0.588	0.725	0.703	L-shaped
YH	infinity	309.6	infinity	0.620	0.763	0.738	L-shaped
PT	infinity	infinity	infinity	0.611	0.753	0.728	L-shaped
TS	694.5	207.6	infinity	0.529	0.680	0.651	L-shaped
XM	infinity	175.1	infinity	0.605	0.749	0.721	L-shaped

## Data Availability

Not applicable.

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
