# Peer review of "Genetic Diversity and Structure Revealed by Genomic Microsatellite Markers of Mytilus unguiculatus in the Coast of China Sea"

_animals, 2023, doi:10.3390/ani13101609_

Round 1

Reviewer 1 Report

Review of manuscript nr: animals-2345407

Mytilus unguiculatus is an offshore shellfish. In the East Asia, it is important component of local aquaculture. Because high demand to products obtained from this species, their populations and stocks are under increasing anthropopressure and effects of intensive aquaculture.

The manuscript:  Genetic diversity and structure revealed by genomic microsat-2 ellite markers of Mytilus unguiculatus in the coast of China Sea that has been sent to me for review is aimed on analysiss of genetic variation inside and between populations of this species.

The results of this study are interesting and potentially useful. They expand some knowledge about relationship between genetic characteristics and geographical distribution of this species, moreover, can be a basis of decisions concerning maintaining genetic diversity of Mytilus unguiculatus.

Generally, the manuscript in version sent to the review is written quite well. The sections: Introduction, Materials and Methods, Results contain most of information that should be given within them. The Discussion and Conclusion require more changes to make them ready for publication. I have not found in this work any serious methodological problem that will require extensive modification of methods used or their supplementation nor faults presentation of results and their interpretation. However, changes necessary to make this manuscript ready for publications are numerous they can be without significant modification of this work.

Summarize, this manuscript requires minor revisions.

General comments:

A.  Whenever the results of these studies are related to the population from which the samples were taken a term “sample” should be replaced by “population/s.” If the samples were taken from groups of animals kept by human for some purposes, then samples should be replaced with term “stock/s.” Please find and correct through the text.

B. Under the term “historical events” the traces of bottleneck or founder effect were investigated. I propose replace term “historical events” it with “bottleneck effect.”

Detailed Comments:

Material and methods

1.        Page 2, line 44. The fragment “wild germplasm resources” should be replaced with “resources of genetic variation in wild populations.”

2.        Page 4, line 116. Fragment “a molecular of variation analysis (AMOVA)”. The term hierarchical analysis of molecular variance (AMOVA) is much more correct and should be used in this place.  

 Results

1.        Page 4, line 134-135. The fragment “Results from the genetic diversity showed sample size (N) of each locus ranged from 46 to 50” is very unclear, therefore must be modified. The symbol “N” is usually used in population genetics as number of investigated individuals taken from given population or from all populations. If under value N the authors inform how many samples amplified successfully, the term “investigated microsatellites were amplified from 46-50 samples and varied between populations.

2.        Page 6, line 149-150. Sentence “In the East China Sea area, the pairwise FST of six samples did not have genetic divergence”. Again, samples should be replaced with word “populations. Moreover, please add ranges of p values of this index. In my opinion entire sentence should be modified as follow “In the six investigated populations from area of East China Sea, their pairwise genetic distances based on fixation index (FST) were not significant (p = (lowest – highest values))

3.        Page 5, line 157. The p should be written in lowercase, and short description of their ranges and interpretation should be given there.   

4.        Page 7, line 174. Ne  should be italicized.

Discussion

A.      In this section the word “samples “should be replaced with “populations”

B.       In this work the likelihood and size of bottlenecks or founder effects were investigated. Why the authors not use term “bottleneck effect”? Wherever the term "historical event" is used in this section I propose replace it with bottleneck effect.

1.        Page 8, line 204-205. Fragment “The results of genetic structure showed that QD sample had a slight genetic variation compared with other samples, which confirmed the geographic distribution of sea areas”. This sentence contains several errors and should be rewritten as follow: “The results of analysis genetic structure and comparison with populations revealed a low level of genetic variation in QS population and confirmed their distribution on areas of sea”.

2.        Page 8, line 215. “Nm” should be corrected to form Nm (N Italicized and  m in down index)

3.        Page 8, line 228-241. The fragment “MtDNA could only detect the genetic diversity of mitochondrial genome. Genetic structure among different samples was obtained based on the mitochondrial genome but not the whole genome in organism. Proportion of mitochondrial genome was tiny compared to the whole genome, which resulted in the different results between the mtDNA and whole genome. When using the molecular markers to analyze the genetic structure among samples, the frequency of gene exchange needed to be considered. With a high frequency of gene exchange, it was difficult to detect the genetic divergence among samples. Therefore, as a narrow scope tool, mtDNA was hard to detect the genetic variation among samples on the high gene exchange frequency. With a wide distribution in the whole genome, researchers can comprehensively access the genetic information of whole genome by using the SSR. When analyzing the genetic structure among samples, the wide distribution of SSR in the whole genome and the polymorphism among samples can do well in obtaining the genetic variation among samples and highlight possible genetic differentiation.” This fragment is example of common knowledge, therefore should be removed.

4.        Page 8, line 242 - 243. The fragment “The results of this study were inclined to M. unguiculatus with a weak and single genetic structure”. This sentence is unclear. It must be modified to show better what the authors are going to tell a reader of this work.

5.        Page 9, line 259 - 260. The fragment “No historical events were significantly obtained in the samples of M. unguiculatus. The inbreeding among samples would be the main factor.” This fragment is unclear and have problems with terminology used: samples instead of populations, inbreeding instead of crossbreeding. I propose to modify this and next sentence to form: “In the investigated populations od M. unguicultas traces of bottleneck effects were not significant, therefore the crossbreeding between populations would be a main factor contributing to actual level of genetic variation”.

6.        Page 9, line 264. The fragment “the genetic lost” should be replaced with “reduction of genetic variation”.

7.            Page 9, line 272 – 273. The fragment “no obvious population events occurred.” does not contain any information. I propose remove this fragment and introduce sentence “The bottleneck effect could affect genetic variation, but in investigated population both presence and size of this reduction is not clear and difficult to assess.”

Tables

1.        Table 2. (Lines of N values). After applying a change proposed in line in Results (Page 4, line 134-135) there is no need show those values in the table. propose remove all lines N lines, the information “investigated microsatellites were amplified from 46-50 samples, and their number varied between populations” introduced to the results section is enough.

2.        Table 5. Ne in title should be italicized.

3.        Table 5. Line 3. Underline below QD and infinity should be removed.

Author Response

Reviewer 1:

Mytilus unguiculatus is an offshore shellfish. In the East Asia, it is important component of local aquaculture. Because high demand to products obtained from this species, their populations and stocks are under increasing anthropopressure and effects of intensive aquaculture.

The manuscript: Genetic diversity and structure revealed by genomic microsat-2 ellite markers of Mytilus unguiculatus in the coast of China Sea that has been sent to me for review is aimed on analysiss of genetic variation inside and between populations of this species.

The results of this study are interesting and potentially useful. They expand some knowledge about relationship between genetic characteristics and geographical distribution of this species, moreover, can be a basis of decisions concerning maintaining genetic diversity of Mytilus unguiculatus.

Generally, the manuscript in version sent to the review is written quite well. The sections: Introduction, Materials and Methods, Results contain most of information that should be given within them. The Discussion and Conclusion require more changes to make them ready for publication. I have not found in this work any serious methodological problem that will require extensive modification of methods used or their supplementation nor faults presentation of results and their interpretation. However, changes necessary to make this manuscript ready for publications are numerous they can be without significant modification of this work.

Summarize, this manuscript requires minor revisions.

Answer: Thank you for your comments on the manuscript. Efforts have been made to clarify the following points and correct the mistakes. The modifications based on the suggestions are explained in further detail below. We have directly revised some minor changes. The corresponding revised manuscript and annotated version with changes highlighted by blue color are submitted.

General comments:

  1. Whenever the results of these studies are related to the population from which the samples were taken a term “sample” should be replaced by “population/s.” If the samples were taken from groups of animals kept by human for some purposes, then samples should be replaced with term “stock/s.” Please find and correct through the text.

Answer: “sample/s” has been changed as “population/s”.

  1. Under the term “historical events” the traces of bottleneck or founder effect were investigated. I propose replace term “historical events” it with “bottleneck effect.”

Answer: “historical events” has been changed as “bottleneck effect”.

Detailed Comments:

Material and methods

  1. Page 2, line 44. The fragment “wild germplasm resources” should be replaced with “resources of genetic variation in wild populations.”

Answer: “wild germplasm resources” has been replaced with “resources of genetic variation in wild populations.

  1. Page 4, line 116. Fragment “a molecular of variation analysis (AMOVA)”. The term hierarchical analysis of molecular variance (AMOVA) is much more correct and should be used in this place.

Answer: this has been modified.

Results

  1. Page 4, line 134-135. The fragment “Results from the genetic diversity showed sample size (N) of each locus ranged from 46 to 50” is very unclear, therefore must be modified. The symbol “N” is usually used in population genetics as number of investigated individuals taken from given population or from all populations. If under value N the authors inform how many samples amplified successfully, the term “investigated microsatellites were amplified from 46-50 samples and varied between populations.

Answer: this has been modified. The corresponding content in the table has also been deleted.

  1. Page 6, line 149-150. Sentence “In the East China Sea area, the pairwise FST of six samples did not have genetic divergence”. Again, samples should be replaced with word “populations. Moreover, please add ranges of p values of this index. In my opinion entire sentence should be modified as follow “In the six investigated populations from area of East China Sea, their pairwise genetic distances based on fixation index (FST) were not significant (p = (lowest – highest values))

Answer: P-values adjusted for multiple comparisons using Bonferroni correction = 0.05/21 = 0.002. This sentence has been modified.

  1. Page 5, line 157. The p should be written in lowercase, and short description of their ranges and interpretation should be given there.

Answer: This has been modified.

  1. Page 7, line 174. Ne should be italicized.

Answer: this has been modified.

Discussion

  1. In this section the word “samples “should be replaced with “populations”

Answer: these have been modified.

  1. In this work the likelihood and size of bottlenecks or founder effects were investigated. Why the authors not use term “bottleneck effect”? Wherever the term "historical event" is used in this section I propose replace it with bottleneck effect.

Answer: this has been modified.

  1. Page 8, line 204-205. Fragment “The results of genetic structure showed that QD sample had a slight genetic variation compared with other samples, which confirmed the geographic distribution of sea areas”. This sentence contains several errors and should be rewritten as follow: “The results of analysis genetic structure and comparison with populations revealed a low level of genetic variation in QS population and confirmed their distribution on areas of sea”.

Answer: the sentence has been rewritten.

  1. Page 8, line 215. “Nm” should be corrected to form Nm (N Italicized and m in down index)

Answer: this has been modified.

  1. Page 8, line 228-241. The fragment “MtDNA could only detect the genetic diversity of mitochondrial genome. Genetic structure among different samples was obtained based on the mitochondrial genome but not the whole genome in organism. Proportion of mitochondrial genome was tiny compared to the whole genome, which resulted in the different results between the mtDNA and whole genome. When using the molecular markers to analyze the genetic structure among samples, the frequency of gene exchange needed to be considered. With a high frequency of gene exchange, it was difficult to detect the genetic divergence among samples. Therefore, as a narrow scope tool, mtDNA was hard to detect the genetic variation among samples on the high gene exchange frequency. With a wide distribution in the whole genome, researchers can comprehensively access the genetic information of whole genome by using the SSR. When analyzing the genetic structure among samples, the wide distribution of SSR in the whole genome and the polymorphism among samples can do well in obtaining the genetic variation among samples and highlight possible genetic differentiation.” This fragment is example of common knowledge, therefore should be removed.

Answer: this fragment has been deleted.

  1. Page 8, line 242 - 243. The fragment “The results of this study were inclined to M. unguiculatus with a weak and single genetic structure”. This sentence is unclear. It must be modified to show better what the authors are going to tell a reader of this work.

Answer: this sentence has been modified.

  1. Page 9, line 259 - 260. The fragment “No historical events were significantly obtained in the samples of M. unguiculatus. The inbreeding among samples would be the main factor.” This fragment is unclear and have problems with terminology used: samples instead of populations, inbreeding instead of crossbreeding. I propose to modify this and next sentence to form: “In the investigated populations od M. unguicultas traces of bottleneck effects were not significant, therefore the crossbreeding between populations would be a main factor contributing to actual level of genetic variation”.

Answer: this sentence has been modified.

  1. Page 9, line 264. The fragment “the genetic lost” should be replaced with “reduction of genetic variation”.

Answer:“the genetic lost” has been replaced with “reduction of genetic variation”.

  1. Page 9, line 272 – 273. The fragment “no obvious population events occurred.” does not contain any information. I propose remove this fragment and introduce sentence “The bottleneck effect could affect genetic variation, but in investigated population both presence and size of this reduction is not clear and difficult to assess.”

Answer: the fragment “no obvious population events occurred.” has been deleted. The sentence “The bottleneck effect could affect genetic variation, but in investigated population both presence and size of this reduction is not clear and difficult to assess.” has been added.

Tables

  1. Table 2. (Lines of N values). After applying a change proposed in line in Results (Page 4, line 134-135) there is no need show those values in the table. propose remove all lines N lines, the information “investigated microsatellites were amplified from 46-50 samples, and their number varied between populations” introduced to the results section is enough.

Answer: the line has been deleted. The corresponding part of the results has also been modified accordingly.

  1. Table 5. Ne in title should be italicized.

Answer: this has been modified.

  1. Table 5. Line 3. Underline below QD and infinity should be removed.

Answer: the underline has been deleted.

Reviewer 2 Report

The authors did a good job reviewing the relevant literature and introducing their study, goals, and objectives.  Materials and methods and results are also presented and illustrated well.  The discussion is  adequate for the most part; however, this reviewer found it difficult to follow the last paragraph of the discussion or the conclusions.

Generally high genetic diversity allows species to adapt to changing environmental conditions and avoid inbreeding. It is not clear whether the authors concluded inbreeding occurrs due to presence of aquaculture species and degradation of habitat?   The conclusion is similarly unclear about the overall message. Perhaps it would be helpful to briefly describe the difference between genetic diversity and genetic differentiation and how the molecular methods varied (if so) in revealing or not revealing differences as they related to natural and aquaculture populations.  The manuscript seemed to fall apart as it neared the end however this may be due to language translation. I could follow this in the early part of the paper (see suggested changes below) but it was more difficult to decipher in the final few paragraphs. 

The manuscript might also be more relevant if the authors briefly state how they believe the data could potentially be used to develop strategies for protecting the species.

Generally, the authors have made significant efforts to present their study in English and much of the paper reads easily; however, as mentioned above, this reviewer had more difficulty interpreting the discussion and conclusion sections.  Here are some suggestions for streamlining the introduction:

Lines 64-70:  ...The purposes of this study were to examine the genetic diversity and population genetic structure of M. unguiculatus using microsatellite markers and to compare the results with studies using mitochondrial molecular markers. Results will aid in defining the geographical distribution patterns of this species, further elucidate the evolution processes, and fill knowledge gaps in related research fields. Moreover, new data will allow fishery resource managers to assess present resources and develop conservation strategies.

Lines 76-80:  ....Fifty individuals were collected and analyzed from each of 7 stations (Table 1, Figure 1). Specimens were collected from the wild environment rather than areas of aquaculture breeding programs. Adductor muscle fragments were removed from each mussel and stored in 95% ethanol and below -20 C until DNA extractions.

Author Response

Reviewer 2:

The authors did a good job reviewing the relevant literature and introducing their study, goals, and objectives. Materials and methods and results are also presented and illustrated well. The discussion is adequate for the most part; however, this reviewer found it difficult to follow the last paragraph of the discussion or the conclusions.

Answer: the last paragraph of the discussion or the conclusions has been modified according to the suggestions of other reviewers.

Generally high genetic diversity allows species to adapt to changing environmental conditions and avoid inbreeding. It is not clear whether the authors concluded inbreeding occurrs due to presence of aquaculture species and degradation of habitat? The conclusion is similarly unclear about the overall message. Perhaps it would be helpful to briefly describe the difference between genetic diversity and genetic differentiation and how the molecular methods varied (if so) in revealing or not revealing differences as they related to natural and aquaculture populations. The manuscript seemed to fall apart as it neared the end however this may be due to language translation. I could follow this in the early part of the paper (see suggested changes below) but it was more difficult to decipher in the final few paragraphs.

Answer: the language expression in the final few paragraphs has been modified.

The manuscript might also be more relevant if the authors briefly state how they believe the data could potentially be used to develop strategies for protecting the species.

Answer: this part has been added in the last part of the paper.

Comments on the Quality of English Language

Generally, the authors have made significant efforts to present their study in English and much of the paper reads easily; however, as mentioned above, this reviewer had more difficulty interpreting the discussion and conclusion sections. Here are some suggestions for streamlining the introduction:

Lines 64-70:  ...The purposes of this study were to examine the genetic diversity and population genetic structure of M. unguiculatus using microsatellite markers and to compare the results with studies using mitochondrial molecular markers. Results will aid in defining the geographical distribution patterns of this species, further elucidate the evolution processes, and fill knowledge gaps in related research fields. Moreover, new data will allow fishery resource managers to assess present resources and develop conservation strategies.

Lines 76-80:  ....Fifty individuals were collected and analyzed from each of 7 stations (Table 1, Figure 1). Specimens were collected from the wild environment rather than areas of aquaculture breeding programs. Adductor muscle fragments were removed from each mussel and stored in 95% ethanol and below -20 C until DNA extractions.

Answer: the introduction has been modified.

Reviewer 3 Report

Overall, this study is promising with significant contributions to understanding the genetic diversity and structure of Mytilus unguiculatus populations along the coast of China Sea. However, improvements are necessary in the language used throughout the paper before it can be considered for final acceptance.

The study employed ten microsatellite loci to investigate the genetic diversity and structure of the populations. The amplification and genotyping results demonstrated high genetic diversity within the species, with weak genetic structure and differentiation observed between the QD sample of the Yellow Sea and other samples of the East China Sea. The structure of the paper is clear, and the population genetic methods used are appropriate. The results support the authors' claims.

I suggest adding a species figure on top of Figure 1.

I agree with the other reviewer's comments regarding the improvement needed in the final paragraphs. My primary concern lies in the language presentation, while I have no significant issues with the methodology or results explanation. For example, I couldn't comprehend the statement "was single and weak" in Line 272. I recommend the authors revise the abstract and conclusions, as the other reviewer also suggested. I apologize for not providing a comprehensive list of language improvements, but given the satisfactory overall science aspect, I believe it is acceptable after language revision.

The language used in the paper overall needs to be improved. For instance, the meaning of the sentence in lines 11-12 is unclear, although it can be understood that QD is genetically differentiated from other populations. Furthermore, the abstract should use present tense instead of past tense to express the study's purpose, objectives, and findings. Overall, the language throughout the paper needs to be improved.

Author Response

Reviewer 3:

Overall, this study is promising with significant contributions to understanding the genetic diversity and structure of Mytilus unguiculatus populations along the coast of China Sea. However, improvements are necessary in the language used throughout the paper before it can be considered for final acceptance.

The study employed ten microsatellite loci to investigate the genetic diversity and structure of the populations. The amplification and genotyping results demonstrated high genetic diversity within the species, with weak genetic structure and differentiation observed between the QD sample of the Yellow Sea and other samples of the East China Sea. The structure of the paper is clear, and the population genetic methods used are appropriate. The results support the authors' claims.

Answer: Thank you for your comments on the manuscript. Efforts have been made to clarify the following points and correct the mistakes. The modifications based on the suggestions are explained in further detail below. The corresponding revised manuscript and annotated version with changes highlighted by blue color are submitted.

I suggest adding a species figure on top of Figure 1.

Answer: Figure 1 has been modified.

I agree with the other reviewer's comments regarding the improvement needed in the final paragraphs. My primary concern lies in the language presentation, while I have no significant issues with the methodology or results explanation. For example, I couldn't comprehend the statement "was single and weak" in Line 272. I recommend the authors revise the abstract and conclusions, as the other reviewer also suggested. I apologize for not providing a comprehensive list of language improvements, but given the satisfactory overall science aspect, I believe it is acceptable after language revision.

Answer: the abstract and conclusions have been revised.

Comments on the Quality of English Language

The language used in the paper overall needs to be improved. For instance, the meaning of the sentence in lines 11-12 is unclear, although it can be understood that QD is genetically differentiated from other populations. Furthermore, the abstract should use present tense instead of past tense to express the study's purpose, objectives, and findings. Overall, the language throughout the paper needs to be improved.

Answer: The language used in the paper has been modified. The tense of abstract has been modified.
